# Possible Role of Pineal and Extra-Pineal Melatonin in Surveillance, Immunity, and First-Line Defense

**DOI:** 10.3390/ijms222212143

**Published:** 2021-11-10

**Authors:** Regina P. Markus, Kassiano S. Sousa, Sanseray da Silveira Cruz-Machado, Pedro A. Fernandes, Zulma S. Ferreira

**Affiliations:** 1Laboratory Chronopharmacology, Department Physiology, Institute Bioscience, University of São Paulo, São Paulo 05508-090, Brazil; sousaks@usp.br (K.S.S.); pacmf@usp.br (P.A.F.); zulmafer@usp.br (Z.S.F.); 2Laboratory of Molecular, Endocrine and Reproductive Pharmacology, Department of Pharmacology, Escola Paulista de Medicina, UNIFESP, São Paulo 04044-020, Brazil; sanseray@hotmail.com

**Keywords:** immune–pineal axis, first-line defense, resident macrophages

## Abstract

Melatonin is a highly conserved molecule found in prokaryotes and eukaryotes that acts as the darkness hormone, translating environmental lighting to the whole body, and as a moderator of innate and acquired defense, migration, and cell proliferation processes. This review evaluates the importance of pineal activity in monitoring PAMPs and DAMPs and in mounting an inflammatory response or innate immune response. Activation of the immune–pineal axis, which coordinates the pro-and anti-inflammatory phases of an innate immune response, is described. PAMPs and DAMPs promote the immediate suppression of melatonin production by the pineal gland, which allows leukocyte migration. Monocyte-derived macrophages, important phagocytes of microbes, and cellular debris produce melatonin locally and thereby initiate the anti-inflammatory phase of the acute inflammatory response. The role of locally produced melatonin in organs that directly contact the external environment, such as the skin and the gastrointestinal and respiratory tracts, is also discussed. In this context, as resident macrophages are self-renewing cells, we explore evidence indicating that, besides avoiding overreaction of the immune system, extra-pineal melatonin has a fundamental role in the homeostasis of organs and tissues.

## 1. Introduction

Health is based on active and constant vigilance to detect pathogens, dead cells, and cellular matrix disruption. In sequence, a quasi-stereotyped reaction kills microorganisms and phagocytes cellular debris, necrotic and apoptotic cells, as well as foreign particles (e.g., diesel particles from air pollution). The migration of neutrophils and monocytes from the blood to the injured area induces a pro-inflammatory phenotype characterized by the release of cytotoxic chemicals (e.g., nitric oxide) and pro-inflammatory cytokines (e.g., TNF). Subsequently, macrophages, the professional phagocytes, clean up and are involved in healing and remodeling. The duration and extent of each phase are tightly regulated to avoid excessive inflammatory or proliferative responses. Thus, surveillance should be constant, and the defense should be strictly controlled to avoid an excess of harmful reactions.

In the last few decades, the understanding of the general mechanistic routes triggered by most aggressors and the specific routes that distinguish a response to a pathogen from a danger/damage-associated molecular pattern has improved significantly. Researchers have disclosed mechanisms involving the integration of organs and systems at the end of the XIX and the beginning of the XX centuries. A good example is presented in the book “Bodily changes in pain, hunger, fear, and rage” [1], which describes the role of sympathetic and parasympathetic systems, as well as adrenal cortisol/corticosterone, on the control of the digestive and circulatory systems. At that time, the integration of isolated responses created the basis for integrative biology. However, the understanding of the molecular pathways inside the cells and molecular signals for the integration of information inside tissues and between organs was developed during the 20th century. Nowadays, based on molecular and cellular mechanisms, we are exploring system biology approach to disclose temporal and spatial integration in physiological, pathophysiological, and pathological conditions.

Here, we will focus on the role of endogenous melatonin as a player in defense responses and how the concept of the immune–pineal axis was developed, focusing on its role in surveillance, defense, the remodeling of tissues, and the restoration of function. We will discuss the switching of melatonin sources from the pineal gland to activated macrophages during the development of an innate immune response and the role of resident macrophages as an extra-pineal source. This new idea offers the potential to tailor the pharmacological use of melatonin and to develop biomarkers based on melatonin sources.

## 2. The Immune–Pineal Axis

### 2.1. Discovery

In the late 1990s, we observed unexpected responses when we compared, for 80 days, the changes in the paw size of mice injected with BCG or a vehicle. BCG induced a significant increase in paw size immediately after injection (hours), followed by a progressive reduction by day 10, and then a second hump that peaked around day 25, followed by a plateau between days 30 and 80 [2]. The time-course of the final plateau, between days 30 and 80, was analyzed using Fourier transform for isolating dominant frequencies [3]. Unexpectedly, the mathematical modeling revealed a circadian rhythm during the chronic phase of the response, with the paw being larger during the day than at night. This change in paw size was due to higher vascular permeability granuloma during the day. Pinealectomy or ablation of the superior cervical ganglia resulted in disrupting the circadian rhythm, which was restored by nighttime oral administration of melatonin [2]. Accordingly, additional studies confirmed that melatonin limits inflammation by reducing the rolling and adhesion of leukocytes to the site of inflammation [4]. Overall, these studies provided a physiological perspective for understanding the role of melatonin in health and as a player in the initial steps of an acute defense response (Figure 1).

In the 1990s, melatonin was extensively studied as an anti-inflammatory drug [6,7]. Melatonin was also shown to regulate immunological organs, such as the thymus, spleen, and adrenal glands [8], and a few studies linked the pineal hormone to the modulation of innate and acquired immune responses [2,4,9,10]. A turning point occurred when we introduced the concept of the immune–pineal axis (IPA), which disclosed how nocturnal pineal melatonin synthesis plays a role in allowing the effective mounting of defense responses. We showed that nocturnal melatonin blocks the rolling and adhesion of leukocytes to the endothelial cells in the healthy state, avoiding the spurious migration of blood cells. The pathogen, danger/damage-associated molecular patterns (PAMPs, DAMPs), and pro-inflammatory cytokines suppress nocturnal melatonin synthesis, favoring the migration of neutrophils and monocytes [11,12,13,14,15]. This concept, proposed in 2007, is now widely accepted due to the integrated view of melatonin’s actions as a physiological regulator of immune responses and a player involved in multiple models of diseases, which has opened perspectives for evidence-based targeting melatonin receptors for therapeutics [11].

Melatonin, synthesized constitutively by resident macrophages or “on-demand” by monocyte-derived macrophages or other immune-competent cells, acts locally, avoiding the overactivation of immune responses, and as a key molecule for directing immunological cells to cope with inflammation. Nowadays, understanding the role of melatonin synthesized by resident macrophages, such as Kupfer cells in the liver and alveolar macrophages in the respiratory tract, opens a new perspective for understanding how melatonin orchestrates a first-line defense, avoiding disturbance of the immune system and homeostasis.

### 2.2. Pineal and Extra-Pineal Orchestrated Melatonin Synthesis

The pineal gland is an active component of the acute defense response. In a healthy state, the increase in nocturnal melatonin in the blood leads to a decrease in adhesion molecules expressed in endothelial cells, reducing the rolling and adhesion of leukocytes to the endothelial layer, and keeping neutrophils and monocytes circulating [5]. Surveillance is a multi-mediated process involving other hormones (cortisol/ corticosterone) and cytokines (IL-10). It is beyond the scope of this review to analyze the mechanisms that mammals developed to detect and respond to pathogens, threats, damage, and stress signals. However, it is relevant to reinforce that, despite being reductants, the mechanisms need to distinguish stimuli and answers according to each body condition.

Despite the great number of studies addressing the anti-inflammatory effect of melatonin, few studies have examined the role of pineal melatonin as a player in surveillance and defense responses. The pineal gland, as a circumventricular organ, monitors, with no barriers, the body and the brain [13]. Pinealocytes and microglia are dispersed throughout the pineal body and the vascular bed, whereas astrocytes marginalize through the sympathetic tract (Figure 2). All three cell types express toll-like and pro-inflammatory cytokine receptors, which transduce extracellular levels of PAMPs, DAMPs, and pro-inflammatory cytokines through the NF-κB pathway [14,15,16]. In addition, the pineal gland presents a high number of receptors known to recognize inflammatory mediators in a rhythmic manner [17]. Nocturnal melatonin synthesis is blocked by pro-inflammatory cytokines (TNF, IL1β), diesel exhausted particles, amyloid β peptide (Aβ), and pathogen-associated molecular patterns from Gram-negative or -positive bacteria, fungi, viruses, and other molecules derived from lipids and toxicants that signalize via NF-κB [5,13,14,18,19,20,21,22].

Melatonin is synthesized by uni- and multicellular organisms from the essential amino acid tryptophan. In mammals, tryptophan is hydroxylated to 5-hydroxytryptophan and then decarboxylated to serotonin (5-hydroxytryptamine). Melatonin is synthesized by two different routes [5,23]. Serotonin might be either methoxylated to 5-methoxytryptamine or acetylated to N-acetylserotonin (NAS), and then the final acetylation of 5-methoxytryptamine or methylation of NAS results in N-acetyl-5methoxy tryptamine (melatonin). In the pineal gland, serotonin is first acetylated by the enzyme serotonin N-acetyltransferase (SNAT, EC 2.3.1.87, also named aryl-alkylamine N-acetyltransferase AA-NAT), and then methylated by the enzyme acetylserotonin O-methyltransferase (ASMT, EC 2.1.14, also known as hydroxy indole-O-methyltransferase) (Figure 2). The conversion of serotonin into NAS by the enzyme SNAT is the key step for synthesizing the pineal hormone at night. Regulation of SNAT coding gene transcription and enzyme phosphorylation is central for the control of melatonin synthesis. The daily melatonin rhythm is driven by nighttime pineal sympathetic input. Beta-adrenoceptors trigger cyclic-AMP/ protein kinase A (PKA) pathway, inducing CREB (cAMP response element-binding protein) and SNAT phosphorylation [24]. The binding of P-CREB to the cAMP response element in the *Aanat* promoter induces its transcription. Phosphorylation at Thr31 enhances up to sevenfold the metabolic process’ efficiency via interaction with the protein 14-3-3, which impairs the ubiquitination and, consequently, the entrance of SNAT into the proteasome [25]. In diurnal animals, gene transcription and protein translation are constitutive. However, the phosphorylation of the enzyme is necessary for impairing proteasome degradation [26]. As a consequence of the differential control of SNAT transcription, plasma melatonin rises in diurnal animals already at the appearance of darkness, whereas it takes 3 to 4 h to peak in nocturnal animals [27].

The IPA is defined by the alternation between pineal (pinealocytes, rhythmic) and extra-pineal (macrophages, non-rhythmic) melatonin synthesis. During the initial phase of the defense response, pineal melatonin synthesis is suppressed. As the response progresses, activated macrophages synthesize melatonin at the injured site [5,22]. The goal of this switch in melatonin sources is to regulate leukocyte migration. At the concentration found in nocturnal plasma, melatonin blocks the rolling and adhesion of leukocytes to the endothelial cells and the synthesis of melatonin by activated macrophages.

Acute inflammation-induced blockage of pineal melatonin synthesis was shown in rodents and anurans challenged with Gram-negative bacteria and TNF [12,13,14,16,21,28]. The IPA is similarly activated in both diurnal and nocturnal species [29,30,31,32,33,34]. Humans responding to acute inflammatory stimuli or disorders accompanied by high TNF also present low nocturnal melatonin production. Women who develop mastitis lose the nocturnal rhythm of melatonin on the first breastfeeding night [35]. The reduction in the melatonin plasma level at nighttime also correlates with high TNF in children with sleep problems [36]. Altogether, both pre-clinical and clinical data highlight the relevance of inflammation in regulating the endocrine response of the pineal gland and reduced hormonal levels of melatonin under inflammatory conditions, regardless of the individual rhythm of activity.

Similarly, immune cells from rodents or humans [10,30,37] can produce melatonin on-demand upon activation, as well as in different physiological [38,39,40] or pathological contexts [41,42,43].

Extra-pineal melatonin also protects against biotic and abiotic primary defense responses that occur independently of the immunological system. The present review aims not to provide a systematic recollection of the constitutive synthesis of melatonin but to compile data supporting the hypothesis that extra-pineal melatonin regulates the acute defense responses and prevents the increased activation of collateral effects. These mechanisms include the negative/positive modulation of gene and protein expression and pathways responsible for pro- and anti-inflammatory phases.

### 2.3. Molecular Mechanisms Involved in the Activation of the Immune–Pineal Axis

The suppression of melatonin synthesis in pinealocytes or its induction in macrophages/microglia relies on the same pivotal pathway: the nuclear translocation of NF-κB (nuclear factor-kappa B) dimers. The NF-κB family is evolutionarily conserved and comprises five subunits (NFκB1 (p50), NFκB2 (p52), RelA (p65), RelB, and cRel) that assemble as homo- or heterodimers and interact with specific DNA kB sequences in target genes, including AA-ANAT/SNAT [5]. The three larger subunits, RelA, RelB, and c-Rel, contain transcription regulatory domains [44]. The p50/RelA dimer induces transcription of the subunit cRel, together with several genes and proteins linked to the acute inflammatory response, such as cytokines (e.g., IL-1β, TNF), adhesion molecules (e.g., ICAM, VCAM, E-selectin), inducible enzymes (COX-2 and iNOS), and some of the acute phase proteins [45]. The cRel/RelA dimer induces the transcription of genes coding for anti-inflammatory and recovery responses [46].

As observed in immunocompetent cells, the pineal gland expresses receptors for several PAMPs and DAMPs (e.g., TLR1, 2, 3, 4, 6, 7, 9, and IL1R1) [17]. Activation of NF-κB in pinealocytes by Gram-negative bacteria components or TNF induces the nuclear translocation of p50/p50 [13,22,41,47]. Studies based on super-shift and chromatin immunoprecipitation show that the *Snat* promoter has two κB elements (nat-κB1 and nat-κB2) [47]. The nat-κB1, which induces the transcription of *Snat*, binds only subunits containing cRel, while nat-κB2, which blocks transcription, binds to other subunits, including p50/p50. These results provided the molecular basis for understanding the opposite responses of pinealocytes and macrophages/microglia to acute inflammatory stimuli and the biological basis underneath the time interval between suppressing the nocturnal pineal melatonin synthesis and the beginning of the synthesis of melatonin by activated macrophages. The local synthesis is induced by inflammatory stimuli, likely framed by the time interval between the induction of cREL transcription and the nuclear translocation of subunits containing cREL (p50/cRel and RelA/cRel). Thus, the shutdown of rhythmic melatonin production is synchronized to provide efficient timing for inflammation management.

The suppression of nocturnal melatonin synthesis by endogenous molecules that signal cellular or matrix disruption does not necessarily depend on the full activation of the IPA. Amyloid-beta peptide Aβ (1–40 and 1–42) and high ATP (mM) block the nocturnal melatonin surge without cytotoxic products. Aβ leads to an eightfold increase in the p50/p50 NF-kB dimer over basal levels, inhibits noradrenaline-induced *Snat* transcription, and increases genes encoding cytokines, TLRs, TLRs adaptors, and downstream signaling proteins [20]. However, Aβ does not increase the transcription of RelA [20]. It is noteworthy that the loss of daily melatonin rhythm is a hallmark of Alzheimer’s disease and occurs much earlier than the clinical symptoms [48,49].

As a co-transmitter of noradrenaline, ATP potentiates melatonin synthesis via the activation of P2Y1 receptors [50,51] (Figure 1). The pineal gland also expresses the low-affinity ATP receptor P2X7, which requires concentrations in the micro- to the millimolar range derived from dead or apoptotic cells [52]. Transcription and translation of the enzyme that converts N-acetylserotonin to melatonin (ASMT) is inhibited by P2X7 activation [53]. Thus, when ATP acts as a DAMP, noradrenaline-induced melatonin synthesis is inhibited, and NAS synthesis is preserved. It is noteworthy that NAS plays a role in mood regulation, stimulates the proliferation of neuroprogenitor cells, and protects retinal photoreceptor cells from light-induced degeneration [54]. Thus, Aβ and ATP reduce nocturnal melatonin via different mechanisms. However, the effect mediated by P2X7 receptors is accompanied by a cytoprotective program. In summary, the pineal gland can recognize and respond to PAMPs and DAMPs depending on the input. Therefore, not all signals that decrease pineal melatonin synthesis fully activate the IPA.

Finally, it is important to consider the role of extra-pineal melatonin, which is synthesized independently of the initiation of defense response. Indeed, these are old/new observations that point to a new structured mechanism for defending multicellular organisms without disturbing the immunological system.

## 3. Melatonin as a First-Line Defense in Organs Exposed to the Environment

The first line of defense avoids the activation of the immunological system. The skin, liver, and lungs are exposed to airborne or ingested microorganisms or toxic particles. Skin [55] and liver [56] melatonin concentrations in healthy rodents and humans are higher than in the serum. Although this comparison has not yet been made for lung melatonin, recently, it was shown that it protects against airborne particulate matter without activating the immune system [22,39].

### 3.1. Skin

Melatonin synthesized in the skin promotes DNA repair and the expression of antioxidant enzymes and regulates mitochondrial function via interaction with cytochrome c and the electron-transport chain. The melatonin concentration in the skin varies in response to environmental stimuli and skin color. The epidermis of African Americans contains the highest concentration of melatonin [57]. The effects of melatonin are mediated by selective, high-affinity G-protein coupled receptors (pM range) and by the direct scavenging of reactive oxygen and nitrogen species (mM range). Skin and liver melatonin reaches mM levels because it is synthesized by two enzymatic pathways [55,56]. Serotonin is acetylated by SNAT (EC 2.3.1.87), the same enzyme expressed in pinealocytes, and by NAT1 (arylamine N-acetyltransferase isoform 1 (EC 2.3.1.5, NAT), an enzyme involved in xenobiont metabolism. This is an ancient route already found in plants and invertebrates [24]. N-acetylserotonin is endogenously produced by normal and malignant melanocytes [57].

To increase its efficiency in scavenging free radicals, melatonin in the skin is metabolized to 2-hydroximelatonin, which leads to two metabolites, AFMK (N(1)-acetyl-N(2)-formyl-5-methoxykynuramine) and AMK (N1-acetyl-5-methoxykynuramine), which are also electron donors [57,58]. The opening of the indole ring occurs via an enzymatic kynurenic pathway and a non-enzymatic ultraviolet B radiation (280–320 nm) interaction [59]. Additionally, AFMK is produced by melanocytes and keratinocytes [57]. In summary, the synthesis of melatonin by human skin has a physiological protective function independent of the activation of the immunological system.

### 3.2. Gastrointestinal Tract

Since 1976, it has been known that the enterochromaffin cells of the gastrointestinal mucosa synthesize melatonin [60]. Melatonin in the gut follows the distribution of serotonin and enterochromaffin cells and modulates local inflammation [61]. Higher concentrations are found in the colon and rectum, while lower concentrations have been found in the jejunum and ileum [62]. Melatonin is synthesized by the gut and its microbiome plays a physiological and defensive role and contributes to circulating hormones. Ingestion of L-tryptophan at daytime, but not at nighttime, increased plasma melatonin in healthy young men [63] and rats [64]. In pinealectomized animals, plasma melatonin increased either at daytime or nighttime [65]. Ligation of the portal vein prevents the rise in blood melatonin after L-tryptophan ingestion, and the elevation of melatonin in portal blood precedes the rise in the bloodstream [64], suggesting that some melatonin reaching the liver is not metabolized and contributes to serum melatonin. Thus, in addition to the pineal gland, melatonin in plasma is also derived from the intestine.

Melatonin from the gut and circulation modulates acute and chronic inflammatory responses. Ischemia/reperfusion-induced acute pancreatitis in rats was attenuated by melatonin or L-tryptophan, converted to melatonin [66]. Higher ASMT expression in enterochromaffin cells, which resulted in an increase in 6-sulfatoximelatonin in the urine and feces, was associated with a lower grade of acute ulcerative proctitis and colitis [67]. The nocturnal increase in melatonin in the bloodstream also plays a role in modulating gut pathophysiology. Exposure of rats to constant darkness alleviated gastritis and colitis [68], and epidemiological reports describe a reduction in symptoms of irritable bowel syndrome at night [69]. Thus, both pineal and extra-pineal melatonin play a role in modulating inflammatory responses in the gut.

Another layer in this complex system is that the diurnal rhythm of the human gut microbiota is entrained by melatonin in humans [70]. This active crosstalk between the human body and its microbiome is the cause of dysbiosis due to changes in daily feeding rhythms and jet lag [70]. In summary, the gastrointestinal tract has its history regarding the melatonergic system, as human cells and the microbiome contribute to melatonin synthesis. In this scenario, melatonin synthesized in the gastrointestinal tract plays local and systemic roles.

### 3.3. Respiratory Tract

Alveolar macrophages, a subset of lung-resident macrophages derived from the yolk sac and fetal liver [71], synthesize melatonin when challenged by biotic and abiotic airborne particles [22]. These macrophages are the first defense barrier against material entering through inhaled air, and they are indispensable for the resolution of pulmonary inflammation [72]. Alveolar macrophages reduce the deleterious effects of apoptotic neutrophils [73], attenuate the inflammatory response induced by the inoculation of large amounts of bacteria [74], and phagocyte diesel exhaust particles, particulate matter, and microorganisms [22,71,75]. During phagocytosis, the increased mucus production facilitates the expelling of cells filled with microorganisms or airborne particles.

Rat lung melatonin at daytime correlates with the Mel-Index [39], an algorithm introduced by our lab to predict the local synthesis of melatonin [41]. The protective role of melatonin synthesized by alveolar macrophages is supported by the fact that the phagocytosis of airborne particles is hampered by the blocking of melatonin receptors [22]. Bioinformatic studies based on microarray data of human alveolar macrophages (GEO data sets) revealed a negative correlation between genes coding for melatonergic function (enzymes for biosynthesis and receptors) and the major enzymes involved in cellular necroptosis (RIPK1, RIPK3, and MLKL), indicating that melatonin synthesized by alveolar macrophages increases alveolar macrophage viability [22]. Moreover, the expression of SNAT and MT1 was positively correlated with the expression of enzymes involved in antioxidant defense (superoxide dismutase, catalase, peroxiredoxins, thioredoxins, and glutathione peroxidase). It is still an open question whether alveolar macrophage melatonin reduces or even prevents the spread of injuring material to the internal organs and the brain [76].

Recently, the COVID-19 pandemic has boosted studies on melatonin as a pharmacological target to control several aspects of the disease and the immunological response triggered by SARS-CoV-2 [77,78,79,80]. All of these studies have focused on the immunological system. However, melatonin synthesized by alveolar macrophages likely coordinates the first line of defense and is an important factor in distinguishing symptomatic from asymptomatic carriers [39].

We constructed gene sets associated with SARS-CoV-2 invasion, replication, and energy expenditure in humans based on a 455 COVID-19 signature using 288 samples of normal human lungs from UCSC Genotype-Tissue Expression RNA-seq [39]. A negative interaction between genes related to virus entry into epithelial AT2 cells and alveolar macrophages and the Mel-Index strongly suggests that high lung melatonin synthesis impairs the entry of the virus into the bloodstream and the triggering of the immune response.

In summary, local melatonin in the skin, gastrointestinal tract, and respiratory tract is part of the first line of defense against invaders and prevents the involvement of the immunological system in an overactive context.

### 3.4. Tissue-Resident Macrophages as a First Defense Line

When we reconstruct the history of melatonin’s involvement in the first line of defense, we consider the organs in direct contact with the environment. Another perspective would evaluate whether there is a relationship between the cells that constitutively produce melatonin in different tissues. Tissue-resident macrophages, cells that sense metabolic changes, tissue damage, and microbial invasion [81], synthesize melatonin constitutively, i.e., independently of IPA activation (Figure 3).

These self-renewing embryonic-derived phagocytes are present in many tissues, and their subtypes and functions are directed toward supplying specific tissues or niches (Figure 3). Resident macrophages of the brain (microglia), bone marrow (αSMA-smooth muscle actin macrophages), peritoneal cavity macrophages, liver (Kupfer cells), spleen (red pulp macrophages), lung (alveolar macrophages and nerve-associated, lung-resident interstitial macrophages), and skin (Langerhans cells) have a common gene signature with yolk sac macrophages and fetal liver [82,83,84,85]. The gastrointestinal tract deviates from this general view because the resident macrophages are derived from blood monocytes [81]. Tissue-resident macrophages are stem cells having the property to proliferate by self-renewal. A wide variety of receptors for recognizing PAMPs and DAMPs and eliciting tissue-specific immune responses can be appreciated at the site of the Immunological Genome Project (http://www.immgen.org, accessed on 30 September 2021). Otherwise, these cells are also part of the first line of defense, a pre-immune response. All resident macrophages studied to date synthesize their own melatonin, which acts in an autocrine or paracrine manner to reduce microorganisms, endogenous harmful particles, or cellular debris without activating harmful mechanisms [5]. Bone marrow a-SMA macrophages maintain hematopoietic stem and progenitor cells in a quiescent state [86]. The daily melatonin rhythm and melatonin synthesized by α-SMA are key elements for maintaining the reserve of hematopoietic stem cells in the long-term stage and reducing the nocturnal differentiation of bone marrow cells [87,88].

**Figure 3 ijms-22-12143-f003:**
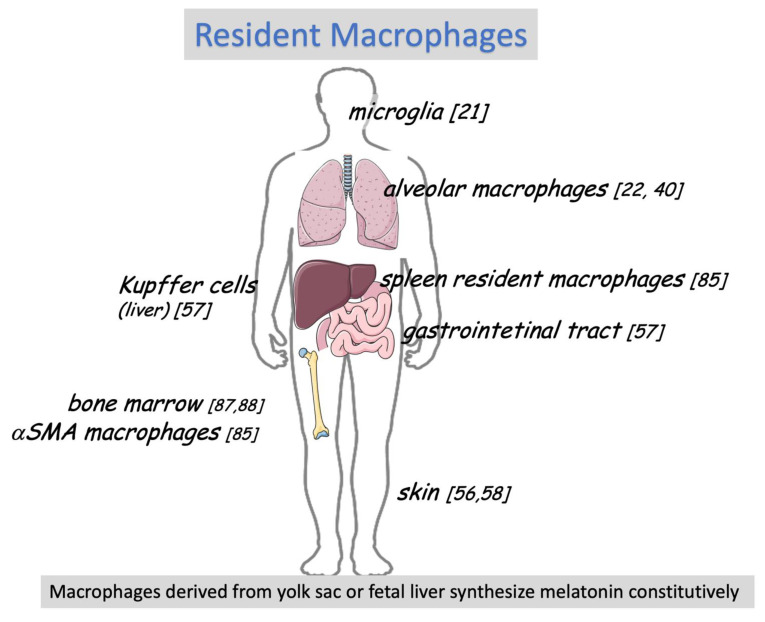
Tissue-resident macrophages and melatonin production—resident macrophages respond to the first defense line against PAMPs and DAMPs. These cells are phenotypically adapted to each tissue and, in most cases, impair the mounting of innate immune responses. All the cells listed above synthesize melatonin. Although the physiological role of tissue-resident macrophages is not yet disclosed in many tissues, in some tissues, as in the bone marrow (BM), it is essential for maintaining a reserve of long-lasting hematopoietic stem cells (see the text).

In conclusion, tissue-resident macrophages are extra-pineal constitutive sources of melatonin, and this multitasking indolamine exerts the function required by each tissue.

## 4. Conclusions and Perspectives

In the early 20th century, McCord and Allen (1917) [89] showed that a highly potent substance present in bovine pineal extracts could lighten the skin of amphibians. Some forty years later, Lerner and colleagues (1959) [90] purified melatonin from bovine pineal extracts of 250,000 cattle, and Julius Axelrod and colleagues proposed in 1961 that melatonin is produced only in the pineal gland. At that time, biological life was viewed as a sum of specialized functions and organs that had to be defined by a specific repertoire of molecules. Because the pineal gland’s function and its hormone were linked to chronobiological events, melatonin was coined as the hormone of darkness. In the latter half of the 20th century, it was recognized that pineal hormones and melatonin synthesized by immunocompetent cells integrate immune system functions and could be a liaison between two highly complex functions: (1) time organization and (2) innate defense. The present text represents a step ahead in integrating the chronobiotic and defense effects of melatonin.

The great relevance of distinguishing the extra-pineal sources of melatonin as those recruited during a defense response and those constitutive is in understanding the multiple roles that melatonin plays when defending an organism and when it supports the recovery from a defense response that relies on oxidative stress. Otherwise, the recovery of the daily melatonin rhythm is not a positive biomarker, as, under chronic inflammation, high adrenal glucocorticoids restore pineal function [9]. Most probably, the nocturnal melatonin rhythm impairs the migration of cells to the lesion site, impairing its resolution.

Understanding where and why melatonin is synthesized and how it contributes to the chain of events involved in defense responses opens new perspectives for tailoring the pharmacological use of melatonergic drugs. We will exemplify the new perspectives focusing on the analgesic effect of melatonin. Pre-clinical and clinical studies showed that melatonin has analgesic and anxiolytic effects (revised by [91]). A clinical trial with 61 healthy subjects showed a significant linear association between serum melatonin and changes in pain threshold and tolerance [92]. Serum melatonin was proportional to sublingual melatonin that was administered in the afternoon. More recently, Kumar and collaborators [93] showed that preanesthetic oral melatonin attenuates anesthetic requirement and hemodynamic responses to intubation. The melatonin group significantly reduced the induction dose of propofol and the number of patients requiring additional fentanyl intraoperatively.

Despite these and other studies supporting an analgesic effect of melatonin, a meta-analysis evaluating postoperative pain and perioperative opioid consumption neither supports nor opposes the assumption that melatonin is a painkiller [94]. Here, we must consider the relevance of activating the IPA by surgical incision or by previous high pro-inflammatory cytokine levels due to pathological conditions. A clinical evaluation of melatonin/ TNF concentration in the plasma of women who underwent elective hysterectomy showed that no nocturnal melatonin was detected on the first night after the surgery and that patients who had higher TNF levels before the surgery had a lower nocturnal melatonin increase after the second night and higher postoperative pain [95]. Finally, here, we point to the relevance of melatonin synthesized by resident macrophages, which represents a defense mechanism similar to those found in many plants and animals and is also linked to chronobiotic hormonal function [96,97].

In summary, melatonin from several sources plays specific and well-defined roles in defense responses. The overview of the sources and actions of endogenous melatonin, given here for the first time, will certainly contribute to the pharmacological use of drugs acting on the melatonergic system. Moreover, extra-pineal or daytime melatonin will be a biomarker to be considered when dealing with acute inflammatory responses.

## Figures and Tables

**Figure 1 ijms-22-12143-f001:**
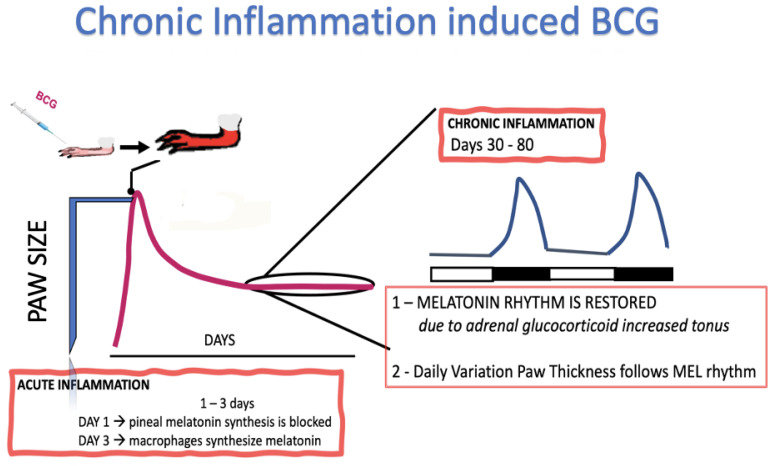
Pineal gland involvement over the time-course of an inflammatory response. BCG injection in a mice paw triggers an acute inflammatory response. A fast increase in the paw size due to swelling and migration of neutrophils and macrophages is followed by the formation of a granuloma. The inhibition of the pineal gland synthesis of melatonin that occurs in the first night is followed by a late recovery of nocturnal melatonin synthesis sustained by the increase in adrenal glucocorticoids. Melatonin rhythm imposes a daily rhythm in paw size [2,4,5].

**Figure 2 ijms-22-12143-f002:**
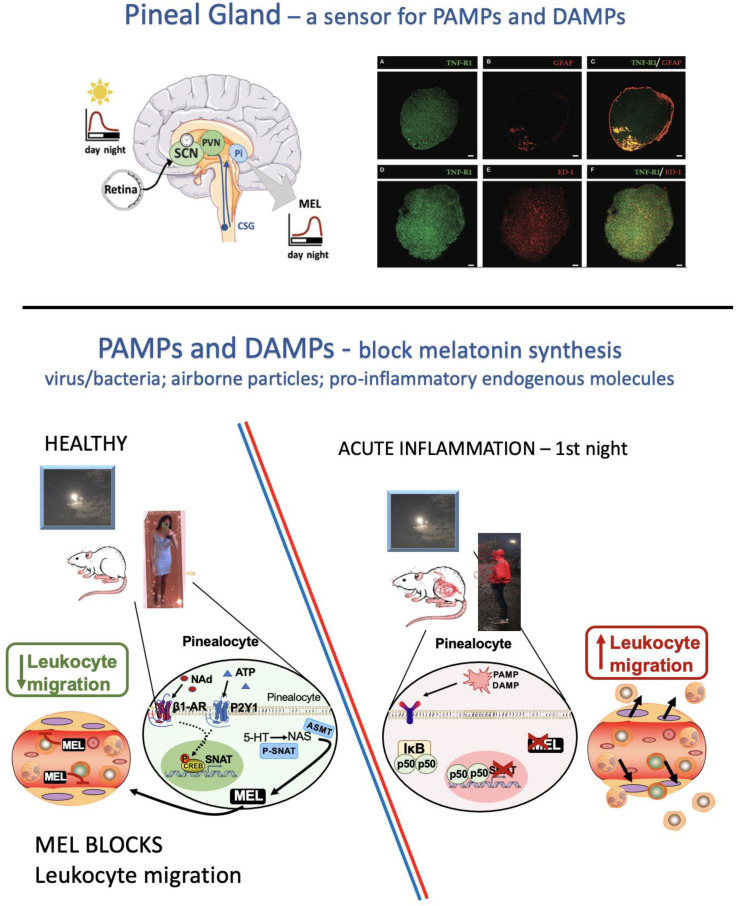
Pineal gland as a reporter of environmental lighting and a sensor of PAMPs and DAMPs. (Upper Panel) The retinohypothalamic tract conveys lighting information to the suprachiasmatic nuclei (SCN). Pineal sympathetic input driven by a polysynaptic pathway via the paraventricular nuclei (PVN) of the hypothalamus and the cervical superior ganglion (CSG) releases noradrenaline (NAd) and ATP. The pineal gland is composed of pinealocytes, astrocytes, and microglia. Astrocytes are localized around the blood vessels, and microglia are dispersed along the pinealocytes. The immunopositive labeling of TNF-R1 throughout the pineal parenchyma is shown in green. The astrocytes (GFAP-labeled, in red) are restricted to the proximal region near the pineal stalk, and most of them express TNFR1 (yellow). Microglia (ED-1-labeled, in red) co-localize with TNF-R1 and pinealocytes also express TNF-R1. Scale bar = 200 μm [16]. (Lower Panel) PAMPs and DAMPs block transcription of SNAT. Left side (healthy conditions): NAd acts on β-adrenoceptors, triggering the adenylyl cyclase/ cAMP/ protein kinase A (PKA) pathway. PKA phosphorylates CREB (cyclic AMP-related responsive element binding protein), which binds to CRE sequences in the *Snat* (serotonin N-acetyltransferase) promoter, inducing the gene’s transcription. The synthesized protein SNAT is activated by PKA phosphorylation. ATP triggers P2Y1 receptors that activate phospholipase C, increasing intracellular Ca2^+^ and potentiating PKA activity. Thus, the conversion of serotonin into N-acetylserotonin (NAS) by P-SNAT occurs at night. NAS is then converted into melatonin by ASMT (acetylserotonin-O methyltransferase). Right side (acute inflammation—1st night): PAMPs and DAMPs activate their receptors in pinealocytes, inducing the nuclear translocation of the NFκB dimer p50/p50, which blocks *Snat* transcription and melatonin synthesis. This is an essential step for mounting acute inflammatory responses at night, as melatonin, via blocking of quinone reductase 2 or activating MT-2 receptors, impairs the migration of leukocytes from the blood to the injured tissue. This 1st-night effect was observed in rodents and humans and is an essential step for mounting an effective innate immune response [13,16,20,21,22].

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
