# Peer review of "Possible Role of Pineal and Extra-Pineal Melatonin in Surveillance, Immunity, and First-Line Defense"

_ijms, 2021, doi:10.3390/ijms222212143_

Round 1
Reviewer 1 Report
The manuscript of Markus et al. deals with the relationship between melatonin and the immune system. It is the work of very experienced authors who have long been involved in research concerning melatonin interaction with inflammatory processes in the body. The presented manuscript is, in principle, a summary of the results of this research group, as it discusses 27 of its own publications. This approach is, in my opinion, fairly acceptable. Still, the whole text often sounds more like a rewritten retrospective lecture with adopted figure panels and some school expressions and explanations (e.g., Fourier's transformation, listing all circumventricular organs when talking only about one).
More importantly, the text is very inattentively written, with many mistakes and often incomprehensible. I understand that some authors send the text to the language proofreading service after the revision. However, this should not be the case with a review paper because the reader cannot rely on the meaning of the presented results but only on the semantic expression of the authors' opinions, which can be distorted due to wrong prepositions, missing verbs and punctuations, or due to the sentences that do not logically follow each other.
As I assume that such an experienced group of authors submitted by mistake an unfinished manuscript, I do not go into detail in the entire text and mention only a few of my objections concerning the first two chapters.
- to make the text more focussed, I suggest removing the second paragraph in the Introduction starting " In the last decade..."
- the next paragraph, starting with: "Here, we will focus..." formulates the goals of the work, and the Introduction should end there.
- the rest of the Introduction could be named ”historical perspective” but has to be rewritten because it is on the border of understanding. Several sentences are difficult to read, but I could not extract the meaning from the sentence starting on line 110 at all.
- line 134 – sentence incomprehensible. The following sentence further frames the goal of the manuscript and belongs to the introduction.
- next paragraph – does the information that PAMPs, DAMPs, and pro-inflammatory cytokines reduce nocturnal melatonin relate? The meaning of the paragraph is difficult to extract; moreover, the info on line 153 seems to be the same as on line 146.
- next paragraph defines melatonin and the principle regulation of its synthesis, which is the main subject of the text for the previous three pages. This paragraph should precede the paragraphs about melatonin function. Furthermore, “SNAT” is not an established abbreviation for aralkylamine N-acetyltransferase. The sentence on line 175 is incomprehensible.
- sentence starting on line 195: Can we talk about “switch” in melatonin production?
- sentence starting on line 204, together with the following sentence, are redundant, repeats the aim of the manuscript instead of conclusive remarks that summarize the meaning of this chapter.
- double citation 55 in figure 3
- figures should be improved as well (different letter types, missing description of cell-types distinguished by colours in Fig. 1)
Author Response
Manuscript IJMS-1382911
Reviewer 1
Authors reply:
Thank you so much for the critics. It was hard to read, but unfortunately, I did a mistake when sent the material for the first time. Certainly, the special conditions that I am working on this year could be blamed, but now, despite being back in the hospital we develop a better way to cope with the situation.
We have extensively revised the text from a scientific point of view and it has been reviewed by friends who are fluent in English.
Before highlighting specific points I would like to put the review into context.
Systematic that guided the review:
1 - show how the idea of the immune-pineal axis evolved from experimental evidence, highlighting the difference between acute inflammatory processes (also known as an innate immune response) and chronic inflammation;
2 - discuss the premises and possibilities of the Immune-Pineal Axis, which acts both in monitoring and in the assembly and regulation of an acute inflammatory response;
3 - highlight the integration of resident macrophages in the melatonergic defense system;
4 - lay the groundwork to be able to investigate the participation of melatonin in the first line of defense, which has the characteristic of not disturbing the immune system.
We understand that the specific points have been detailed only as examples. They have been numbered and commented on below:
- to make the text more focussed, I suggest removing the second paragraph in the Introduction starting " In the last decade..."
- the next paragraph, starting with: "Here, we will focus..." formulates the goals of the work, and the Introduction should end there.
ANSWER:
OK, we agree with the suggestion. The last paragraph of the Introduction now reads:
"Here we will focus on the role of endogenous melatonin as a player of defense responses and how the concept of the immune-pineal axis was developed, focusing on its role in surveillance, defense, remodeling of tissues, and restoration of functions. We will discuss the switching of melatonin sources from the pineal gland to activated macrophages during the development of an innate immune response and the role of resident macrophages as an extra-pineal source. This new idea opens the perspective to tailor the pharmacological use of melatonin and to develop biomarkers based on melatonin sources."
3. The rest of the Introduction could be named ”historical perspective” but has to be rewritten because it is on the border of understanding. Several sentences are difficult to read, but I could not extract the meaning from the sentence starting on line 110 at all.
ANSWER:
Thanks for the suggestion. We now present the two pillars that we consider to be the basis for the action of melatonin in mammalian defense. 1 - integration between chronobiotic and immunological functions represented by the immune-pineal axis; 2 - the first-line defense that does not even alert the immune system of the disturbance in question, represented by resident macrophages.
4. line 134 – sentence incomprehensible. The following sentence further frames the goal of the manuscript and belongs to the introduction. 
ANSWER: WITHDRAWL.
5. next paragraph – does the information that PAMPs, DAMPs, and pro-inflammatory cytokines reduce nocturnal melatonin relate? The meaning of the paragraph is difficult to extract; moreover, the info on line 153 seems to be the same as on line 146:
ANSWER:
All the information was rephrased. The main idea is showing that “the suppression of nocturnal melatonin synthesis by endogenous molecules that signal cellular or matrix disruption does not necessarily depend on the full activation of the IPA” (lines 240-261). This is a new approach to understand how endogenous signals of danger or biomarkers of pathologies integrate pineal function as pathophysiological or pathological responses.
6. next paragraph defines melatonin and the principle regulation of its synthesis, which is the main subject of the text for the previous three pages. This paragraph should precede the paragraphs about melatonin function. Furthermore, “SNAT” is not an established abbreviation for aryalkylamine N-acetyltransferase. The sentence on line 175 is incomprehensible.
ANSWER:
Indeed, SNAT is the abbreviation accepted for the protein (https://www.uniprot.org/uniprot/Q16613), while for the gene some sites still use AA-NAT (https://www.genecards.org/cgi-bin/carddisp.pl?gene=AANAT). Now we mention the two names but still did not explore the issue. For years we used AA-NAT, however, the name serotonin N-acetyltransferase sounds easier to those not so familiar with chemical formulae.
Sentence on line 174 – old manuscript:
“As a consequence of different control of SNAT transcription, plasma melatonin rises in diurnal animals already at the entrance of darkness, while in nocturnal animals it 3 to 4 hours to reach the peak (Borjigi and Liu, 2008).”
Thanks for calling our attention to this important sentence. In the revised manuscript it is in line 157 and reads “As a consequence of the differential control of SNAT encoding gene transcription, plasma melatonin rises in diurnal animals already at the entrance of darkness, whereas it takes 3 to 4 hours to peak in nocturnal animals [27].”
7. sentence starting on line 195: Can we talk about “switch” in melatonin production?
ANSWER:
The basis of the immune-pineal axis is the switch between the pineal and activated macrophages melatonin synthesis. The concept presented in 2007 (Markus et al, 2007) was accepted by several researchers in the area. The biochemical basis is the activation of the NF-kB pathway both in pinealocytes and activated macrophages. In pinealocytes, the nuclear translocation of the dimers p50/p50 results in suppression of Aa-nat transcription, whilst in macrophages cRel is synthesized. The dimers cRel/RelA and RelA/p50 interact with the promoter of Aa-nat inducing its transcription. We include the idea that this sequence of events organizes the migration of leukocytes induced by an acute injury (microorganisms, via toll-receptors or others, such as a surgical incision, that will raise TNF).
8. sentence starting on line 204, together with the following sentence, are redundant, repeats the aim of the manuscript instead of conclusive remarks that summarize the meaning of this chapter.
ANSWER:
Thanks for the observation.
9. double citation 55 in figure –
ANSWER:
All the figures were edited.
10. figures should be improved as well (different letter types, missing description of cell-types distinguished by colors in Fig. 1).
ANSWER: In figure 1 the real records are shown – thus we cannot change the colors.
Reviewer 2 Report
This article provides a detailed overview of the role of melatonin in surveillance, immune and first line defense.
Unfortunately, studies reported were done almost exclusively in nocturnal species ie species that are highly active at night, the only time when pineal melatonin is actively synthesized and released. Little consideration is given in the text with respect to diurnal species, those in which most activity conicides with little or no circulatory melatonin (that comes almost eclusively from the pineal gland). Is the immune-pineal (axis) interaction present in diurnal species and if so how is it expressed. Clearly there is a strong role of melatonin in first line defense in diurnal species, but does suppression of pineal activity have any role?
Author Response
Reviewer 2
This article provides a detailed overview of the role of melatonin in surveillance, immune and first line defense. Unfortunately, studies reported were done almost exclusively in nocturnal species ie species that are highly active at night, the only time when pineal melatonin is actively synthesized and released. Little consideration is given in the text with respect to diurnal species, those in which most activity coincides with little or no circulatory melatonin (that comes almost exclusively from the pineal gland). Is the immune-pineal (axis) interaction present in diurnal species and if so how is it expressed. Clearly there is a strong role of melatonin in first line defense in diurnal species, but does suppression of pineal activity have any role?
Authors reply:
We appreciate the comments and suggestions.
Considering the question “Is the immune-pineal (axis) interaction present in diurnal species and if so, how is it expressed?”
ANSWER:
We thank you so much for the observation and clearly stated that in both diurnal and nocturnal animals the activation of the immune-pineal axis was observed.
Lines 141-164 – here we highlighted that “As a consequence of the differential control of SNAT transcription, plasma melatonin rises in diurnal animals already at the entrance of darkness, whereas it takes 3 to 4 hours to peak in nocturnal animals [27]. “The IPA is similarly activated in both diurnal and nocturnal species [30–35]” Lines 191 – 203.
Round 2
Reviewer 1 Report
I apologize for the stark evaluation of the original text. It arose from a certain disappointment of expectations. However, this expectation was met by the new version. The article is very interesting, and the point of view on the relationship of melatonin to the immune system I consider as original. I only have a minor objection related to Figure 2 - I do not understand the meaning of the pictograms in the blue box and the purpose of both figures - blue and red. Maybe it could be better explained in the image legend or directly in the image.
Author Response
We would like to thank the Reviewer for the questioning. Indeed, this manuscript was written in special circumstances.
We clarified the doubts including in the legend the information that, regarding surveillance and acute inflammatory response, melatonin has the same effect in diurnal and nocturnal animals, i.e., rodents and humans.
Reviewer 2 Report
The authors have described the concept of the immune-pineal system fully and have adequately documented it.
Author Response
We would like to thank the Reviewer.
We included in the legend the information that, regarding surveillance and acute inflammatory response, melatonin has the same effect in diurnal and nocturnal animals, i.e., rodents and humans.